# New Hybrid Spikelet Sterility Gene Found in Interspecific Cross between *Oryza sativa* and *O. meridionalis*

**DOI:** 10.3390/plants11030378

**Published:** 2022-01-29

**Authors:** Katsuyuki Ichitani, Daiki Toyomoto, Masato Uemura, Kentaro Monda, Makoto Ichikawa, Robert Henry, Tadashi Sato, Satoru Taura, Ryuji Ishikawa

**Affiliations:** 1United Graduate School of Agricultural Sciences, Kagoshima University, 1-21-24 Korimoto, Kagoshima 890-0065, Kagoshima, Japan; 2Faculty of Agriculture, Kagoshima University, 1-21-24 Korimoto, Kagoshima 890-0065, Kagoshima, Japan; 3Graduate School of Agriculture, Forestry and Fisheries, Kagoshima University, 1-21-24 Korimoto, Kagoshima 890-0065, Kagoshima, Japan; 4Queensland Alliance for Agriculture and Food Innovation, University of Queensland, Brisbane, QLD 4072, Australia; robert.henry@uq.edu.au; 5Graduate School of Life Science, Tohoku University, Sendai 980-8577, Miyagi, Japan; tadashi-satoh@est.hi-ho.ne.jp; 6Institute of Gene Research, Kagoshima University, 1-21-24 Korimoto, Kagoshima 890-0065, Kagoshima, Japan; taura@gene4.agri.kagoshima-u.ac.jp; 7Faculty of Agriculture and Life Science, Hirosaki University, Hirosaki 036-8561, Aomori, Japan; ishikawa@hirosaki-u.ac.jp

**Keywords:** reproductive barrier, hybrid sterility, egg-killer, wild rice, *O*. *meridionalis*, *O*. *sativa*

## Abstract

Various kinds of reproductive barriers have been reported in intraspecific and interspecific crosses between the AA genome *Oryza* species, to which Asian rice (*O. sativa*) and African rice (*O. glaberrima*) belong. A hybrid seed sterility phenomenon was found in the progeny of the cross between *O. sativa* and *O. meridionalis*, which is found in Northern Australia and Indonesia and has diverged from the other AA genome species. This phenomenon could be explained by an egg-killer model. Linkage analysis using DNA markers showed that the causal gene was located on the distal end of chromosome 1. Because no known egg-killer gene was located in that chromosomal region, this gene was named *HYBRID SPIKELET STERILITY 57* (abbreviated form, *S57*). In heterozygotes, the eggs carrying the *sativa* allele are killed, causing semi-sterility. This killer system works incompletely: some eggs carrying the *sativa* allele survive and can be fertilized. The distribution of alleles in wild populations of *O. meridionalis* was discussed from the perspective of genetic differentiation of populations.

## 1. Introduction

Reproductive barriers have been observed in the intraspecific and interspecific cross between the AA genome *Oryza* species, to which Asian rice (*O. sativa*) and African rice (*O. glaberrima*) belong. We identified some causal genes of these reproductive barriers, such as hybrid weakness [1,2,3] and hybrid chlorosis [4] in *Oryza sativa*. These barriers are obstacles to the genetic improvement of rice, on which more than half of all people depend for staple food. Reproductive barriers have also been studied in the context of differentiation and speciation [5,6,7]. Detailed studies on reproductive barriers contribute to both rice breeding and evolutionary biology [8,9]. Among reproductive barriers, hybrid seed (spikelet) sterility has been intensively studied because it is related to hybrid rice breeding. Asian rice *O. sativa* comprises two major subspecific groups, *indi*ca and *japonica*. The hybrid between *indica* and *japonica* shows hybrid vigor, but also shows hybrid seed sterility [10]. The major gene causing hybrid seed sterility is *S5* [11]. The hybrid seed sterility caused by the *S5* gene is explained by an egg-killer model: In heterozygotes of *S5*-i derived from indica and *S5*-j derived from japonica, *S5*-i and *S5*-j respectively act as killer allele and abortive allele, leading to semi-sterility, with about half of the seeds being sterile. Other genes, such as *S7* and *S9*, also follow the same egg-killer model [12,13]. The interspecific crosses between *O. sativa* and the African rice *O. glaberrima* show hybrid vigor but are highly sterile both in pollen and seeds [14]. Some causes of this hybrid sterility are explained by a gametic eliminator model [14,15,16]: In a heterozygous state of killer allele and abortive allele, both the eggs and pollen carrying the abortive allele are killed in the gametic eliminator model, causing semi-sterility in both spikelet and pollen. In the selfed progeny of the egg-killer model, the ratio of homozygotes of killer allele and heterozygotes are expected to be 1:1, and no homozygotes of the abortive allele are expected to appear. On the other hand, in the selfed progeny of the gametic eliminator model, only homozygotes of the killer allele are expected to appear. Mapping and characterization of the causal genes of hybrid seed and pollen sterility could contribute to hybrid rice breeding.

*O. meridionalis* is an AA genome *Oryza* species from Northern Australia and Indonesia [17]. Several types of molecular data indicate that this species has diverged from the other AA genome species and that it is distantly related to them [18,19]. This divergence is reflected by the low pollen fertility of the hybrid between *O. meridionalis* and the other AA genome species [20,21,22]. Li et al. [23] identified five hybrid pollen sterility genes, all of which follow the pollen-killer model [15,16]. No genes conferring hybrid seed sterility have been reported in the cross between *O. meridionalis* and the other AA genome species before our recent study [24], in which we reported seed abortion after fertilization in the progeny from the cross between *O. sativa* and *O. meridionalis* was controlled by a *SEED DEVELOPMENT 1* (*SDV1*) gene and a *SEED DEVELOPMENT 2* (*SDV2*) gene. This gene model is not an egg-killer model but a duplication and loss of a functional gene for seed development—a new finding in rice genetics. However, seed sterility is not completely explained by these genes in the cross between an *O. meridionais* strain Jpn2 and *O. sativa*. In the present study, we report that an egg-killer gene also contributes to the sterility in the above cross combination. Judging from the gene location and gene action, this gene is thought to be a new finding in rice genetics.

## 2. Results

### 2.1. Finding of a Hybrid Sterility Phenomenon

The BC_4_F_2_ population using Jpn2 as a donor parent and an *Oryza sativa* cultivar Taichung 65 (T65) as the recurrent parent showed diverse seed fertility, which was not explained only by the *SDV1* gene [24].

To remove the genetic noise, we backcrossed further to produce a BC_5_F_2_. However, the situation was not improved (Figure 1). The *SDV1* gene was linked closely to *HD1*, a major heading-time gene [24,25]. Therefore, early heading plants for which days to heading ranged from 98 to 106 were expected to be homozygotes of the *Sdv1-s* allele, the functional allele for seed development, and to be segregating for the other gene(s) causing seed fertility. Ten plants varying in seed fertility were selected, and their selfed progenies were checked for seed fertility (Table 1). All the plants in the BC_5_F_3_ lines headed as early as the recurrent parent T65, suggesting that they were homozygotes of T65 at the *HD1* locus (data not shown). Three BC_5_F_3_ lines derived from plants with fertility ranging from 80.3% to 88.7% were skewed toward high fertility. Three BC_5_F_3_ lines derived from plants with fertility ranging from 57.3% to 79.3% showed a wider range of seed fertility. The other four BC_5_F_3_ lines derived from plants with fertility ranging from 44.0% to 54.0% showed bimodal seed fertility with one peak around 50% and the other peak around 90%. Among them, BC_5_F_3_ line 8 showed clear bimodal distributions comprising 16 semi-sterile plants with fertility ranging from 0.45 to 0.65 and 18 fertile plants with fertility ranging from 0.85 to 1.00. 

Bulked DNA composed of 10 randomly chosen plants per BC_5_F_3_ line were subjected to genotyping of 76 DNA markers covering the whole 12 rice chromosomes, with the result that the genotype of S13374 (Table 2), located on the distal end of the short arm of chromosome 1, showed four homozygotes of the Jpn2 allele and six heterozygotes, though these lines underwent backcross five times using T65 as the recurrent parent. All the homozygotes of Jpn2 were derived from BC_5_F_2_ plants with high seed fertility, and all the heterozygotes were derived from BC_5_F_2_ plants with semi-sterility. These experimental results could be explained by the hypothesis that eggs carrying a gene linked with S13374 from T65 are selectively eliminated during egg development in heterozygotes, following an egg-killer model. Deformed pollen and weakly stained pollen were observed in some low seed-fertility plants. However, the degree of deformation and stain varied highly and was difficult to quantify (data not shown). All plants of BC_5_F_3_ line 8 showed high pollen fertility (more than 90%). Therefore, the cause of seed fertility could be confined to the egg. A BC_5_F_4_ line derived from a semi-sterile BC_5_F_3_ line 8 was used for further analysis.

### 2.2. Linkage Analysis of the Causal Gene of the Hybrid Sterility

The distribution of the BC_5_F_4_ line showed a clear bimodal distribution of seed fertility (Figure 2). When divided by the dotted line in Figure 2, the ratio of the semi-sterile group and fertile group was 37:39, very close to 1:1. Most semi-sterile plants were heterozygotes, and most fertile plants were homozygotes of the Jpn2 allele at S13374. These experimental results could be explained by the above hypothesis. 

Then, we examined the BC_5_F_5_ lines shown in Table 3. We designed five PCR-based DNA markers located close to S13374 (Table 2) and checked the bulk DNA of each BC_5_F_5_ line for the genotypes of these DNA markers. Then we genotyped the six DNA markers in Table 2 of a total of 506 plants composed of BC_5_F_5_ line 1–9 to find recombinants between the DNA markers. 

The seed fertility and genotypes of the DNA markers of the recombinants suggested that the causal gene of semi-sterility is located between KGC1_1.24 and KGC1_1.67 (Table 4). Heterozygotes of KGC1_1.24 showed semi-sterility and homozygotes of Jpn2 showed high fertility, except that line 7 individual 61 (hereafter abbreviated as 7-61) showed high fertility, though it was heterozygous at KGC1_1.24. Then 15 non-recombinants between the two markers of each BC_5_F_5_ line were evaluated for seed fertility. Hence, all data support the evidence that the causal gene was located close to KGC1_1.24 (Table 3). 

To confirm the location of the causal gene, the progeny of the above recombinants were genotyped for KGC1_1.24 or KGC1_1.67. Segregation distortion occurred in the upper three BC_5_F_6_ lines while it did not in the lower six BC_5_F_6_ lines in Table 4, which is consistent with the seed fertility of the previous generation. The genotypes of the causal sterility gene were the same as those of KGC1_1.24 for all the recombinants in Table 4 except 7-61 in that the genotype was the same as that of KGC1_1.67. About 30 plants of BC_5_F_6_ lines from 3-21, 7-61 and 3-57 were checked for seed fertility (Table 4). The distribution of seed fertility of each line was consistent with the inferred genotypes of the causal gene.

Based on the inference that the genotype of the causal gene was the same as KGC1_1.24 except 7-61 (Table 4), the haplotypes of 506 plants composed of BC_5_F_5_ line 1-9 (Table 3) surrounding the causal gene were classified, as shown in Table 5. As will be described in the discussion section, the causal gene was named *S57*. A total of 17 homozygotes of the T65 allele confirmed that the selective elimination of the T65 allele during egg development is incomplete. The linkage map of the causal gene was constructed based on Table 6 and genotypes of S13374, KGC1_0.61, KGC1_2.24 and KGC1_2.50 (Figure 3). The *S57* gene was closely linked with KGC1_1.24, with a genetic distance of 0.1cM calculated by the Kosambi function.

### 2.3. Confirmation of Egg-Killer Phenomenon by Testcross

Apart from the BC_5_F_5_ lines for linkage analysis of the *S57* gene, we separately planted about 15 plants each from the BC_5_F_5_ line 1-9. The three plants heterozygous at S13374 were reciprocally backcrossed with the recurrent parent to produce BC_6_F_1_ generation and selfed to produce BC_5_F_6_ generation. When T65 was used as a pollen donor, the genotype ratio of KGC1_1.24 deviated significantly from the expected ratio of 1:1 (Table 6). The BC_5_F_6_ generation also showed segregation distortion. On the other hand, when T65 was used as an egg donor, the genotype ratio of KGC1_1.24 of BC_6_F_1_ generation fitted the expected ratio. This result confirms that S57 is an egg-killer gene.

### 2.4. Distribution of the Egg-Killer Gene

In our previous paper [24], seed sterility caused by genes other than *SDV1* was observed only in the cross with Jpn2. We inferred the genotype of the causal sterility gene in this study of another *O. meridionalis* strain W1297 and an Australian *O. rufipogon* strain Jpn1 [24] by the seed fertility pollinated by T65 pollen of the BC_1_F_1_, BC_2_F_1_ and BC_3_F_1_ generation. When using Jpn1 as the donor parent, average seed fertility remained above 69%, indicating that Jpn1 does not carry a causal gene of semi-sterility (Figure 4). When W1297 was used as donor parent, average seed fertility elevated as backcross times increased, and in the BC_3_F_1_ generation, the average seed fertility was 71% with the seed fertility of few plants less than 50%. This is in contrast to the case of Jpn2: where seed fertility of BC_1_F_1_ was 26%, and that of BC_3_F_1_ was 49% with 29 BC_3_F_1_plants showing seed fertility less than 50%. The *S57* gene found in Jpn2 almost completely eliminates the egg carrying the T65 allele. Therefore, when backcrossed using T65 as the pollen donor, the Jpn2 allele tends to be transmitted preferentially and remains the cause of semi-sterility. Because no similar tendency as Jpn2 was observed in W1297, W1297 is thought to not carry a causal gene for hybrid seed sterility.

## 3. Discussion

In this study, seed semi-sterility observed in the backcross progeny using an *O. meridionalis* strain Jpn2 as donor parent and T65 as a recurrent parent was genetically analyzed. A gene from Jpn2 was the cause of semi-sterility. As shown in Figure 5, in heterozygous form, eggs carrying the T65 allele were incompletely sterilized. This gene exerts no effect on pollen sterilization. Therefore, this phenomenon can be explained by an egg-killer gene model. Linkage analysis indicated that this gene is located on the distal end of the short arm of chromosome 1 (Figure 3). Genes with similar functions were observed in intersubspecific crosses in *Oryza sativa*, such as *S5* [11], *S7* [12] and *S9* [13]. According to [27], in the cross between *O. sativa* and *O. meridionalis*, only pollen-killer genes have been reported. Li et al. [23] performed linkage analysis of five pollen-killer genes: *S51* located on chromosome 1, *S52* and *S53* located on chromosome 2, *S54* and *S55* on chromosome 7. Yu et al. [28] also identified the pollen-killer gene *qSHM7* in the cross between *O. sativa* and *O. meridionalis* and located it on the same chromosomal location as *S55*. Furthermore, *O. meridionalis* alleles behaved as pollen-killer alleles at the *S51* and *S54* loci and behaved as abortive alleles at the *S52*, *S53* and *S55* (*qHMS7*) loci. To our knowledge, there have been no reports of egg-killer genes in this chromosomal region, and no report of those found in the cross between *O. sativa* and *O. meridionalis*. Two egg-killer genes were reported to be located on chromosome 1: *ESA* was detected in the cross between *O. sativa* and *O. rufipogon*. This gene was tightly linked with RM24, located at 18.9 Mb on chromosome 1 of IRGSP 1.0 pseudomolecule [29]. *S40* was detected in the cross between *O. sativa* and *O. longistaminata*. This gene was tightly linked with RM575, located at 8.1 Mb [30]. Therefore, the egg-killer gene in this study is a new gene. 

In the history of rice gene nomenclature, S with digits have been applied to the gene symbols of both pollen sterility genes and egg sterility genes. According to a database of rice, Oryzabase (Oryzabase.Available online: https://shigen.nig.ac.jp/rice/oryzabase/locale/change?lang=en accessed on 31 December 2021) [31], and a recent review on hybrid sterility by Li et al. [27], *S56(t)* is the latest gene symbol, which was found in the cross between *O. sativa* and *O. glumaepatula*, and caused pollen semi-sterility [32]. Therefore, we name the gene found in this study as *HYBRID SPIKELET STERILITY 57*, with the gene symbol of *S57*, following Yoshimura and Nagato [33] and McCouch and CGSNL [34]. In the intraspecific crosses among *O. sativa*, there have been no reports of hybrid spikelet sterility phenomena caused by a gene located on the distal end of the short arm of chromosome 1. Therefore, all *O*. *sativa* is thought to share the same allele found in T65. This allele was called *S57-s* (s for *sativa*). The allele of Jpn2 was not shared by another *O. meridionalis* strain, W1297. This allele was called *S57-j* (j for Jpn2). 

S57 is located in the 430.6 kb chromosomal region encompassed by the two DNA markers KGC1_1.24 and KGC1_.67. We are undertaking fine-mapping of the S57 gene to identify the causal DNA sequence diversifying the function of the S57 gene. According to Rice Genome Annotation Project (Rice Genome Annotation Project. Available online: http://rice.uga.edu/index.shtml accessed on 15 January 2022) [35], 69 genes with the gene name LOC_Os01g0**** are located on that region in *O. sativa* cultivar Nipponbare genome IRGSP 1.0 pseudomolecule [35]. We are undertaking fine-mapping of the S57 gene to identify the causal DNA sequence diversifying the function of the S57 gene. In the case of the *S1* gene, the corresponding DNA sequence of killer allele in *O. glaberrima* is not present only in *O. sativa* genome [15]. After delimiting the candidate chromosomal region, we will search for the candidate gene in both *O. sativa* and *O. meridionalis* genomes. The project of sequencing the Jpn2 genome is underway to uncover *S57*, *SDV1*, *SDV2* and long grain gene(s) from Jpn2 [36].

In a review by Calvo-Baltanás et al. [37] on hybrid incompatibility, genetic and molecular studies of hybrid incompatibility in numerous plant species revealed that such self-destructing symptoms in most cases are attributed to autoimmunity: plant immune responses are inadvertently activated in the absence of pathogenic invasion. Most of the hybrid incompatibility, such as hybrid weakness and hybrid necrosis, is explained by a conflict involving a member of the major plant immune receptor family, the nucleotide-binding domain and leucine-rich repeat-containing protein (NLR; formerly known as NBS-LRR). NLR genes are associated with disease resistance traits. According to [37], hybrid sterility can be explained in the same context. The *S5* gene encodes a disease resistance-related aspartic protease (AP) [11]. Although the AP family has not yet been investigated in line with NLR activity, members of this family are involved in defense responses with activated salicylic acid in addition to pollen and ovule development in multiple plant species. Furthermore, the *S5* gene is actually a gene complex of three open reading frames, ORF3, ORF4 and ORF5 [11]. ORF5 encodes AP. AP from *indica S5* allele behaves as a killer. Because products from *indica* ORF3 behave as the protector from killer AP, eggs carrying *indica S5* alleles are viable. A pollen-killer gene *qHMS7* is actually composed of two tightly linked genes, ORF2 and ORF3 [28]. ORF2 encodes a ribosome-inactivating protein (RIP) domain-containing protein, which behaves as a toxic genetic element that aborts pollen in a sporophytic manner, whereas ORF3 is predicted to encode homologous, grass family-specific proteins with a mitochondrial targeting signal at the N terminus, which behaves as an antidote that protects pollen in a gametophytic manner. RIPs are toxic RNA N-glycosidases that affect translation processes and have been implicated in apoptotic pathways in mammalian cells and antiviral, antifungal and insecticidal activities in plants [38]. The above two abortion models involve proteins related to plant immunity. Therefore, the S57 egg-killer system might evolve from genes conferring plant immunity and be composed of tightly linked plural genes. Fine mapping of this gene will test the universality of killer-protector systems reported earlier.

According to [27], the *S53* gene [23], a pollen-killer gene found in the cross between *O. sativa* and *O. meridiolis*, is located on the same location as other pollen-killer genes, *S22A*, *S22B* [39] and *S29* [40] on the short arm of chromosome 2. In the cross between *O. sativa* and *O. meridiolis*, *O. sativa* allele behaves as pollen-killer and *O. meridionalis* allele as abortive allele at *S53* locus. In the cross between *O. sativa* and *O. glumaepatula*, the *O. sativa* allele behaves as pollen-killer and *O. glumaepatula* allele as abortive allele at S22A and S22B loci. In the cross between *O. sativa* and *O. glaberrima*, the *O. glaberrima* allele behaves as pollen-killer and *O. sativa* allele as abortive allele at *S29* locus. If *S53*, *S29* and *S22A* or *S22B* are located on the same locus, the hierarchy of tentative allelic interaction would be that *O. glaberrima* carries the strongest egg-killer allele, followed by *O. sativa*, and *O. meridionalis* and *O. glumaepatula* carry abortive alleles. These kinds of allelic hierarchy could be detected by testcross: The progeny of homozygotes of *S57-j* of BC_5_F_6_ will be the tester lines for the allele of *S57*. Introgression lines of AA genome wild rice chromosomal segments under T65 genetic background were developed [41,42]. The testcross with homozygotes of *S57-j* and intercrosses among introgression lines carrying the introgressed chromosomal segments of the distal end of chromosome 1 would clarify the hierarchy of alleles at the *S57* locus. 

Lam et al. [22] reported that the hybrid between Jpn2 and W1297, that between Jpn2 and another *O. meridionalis* strain W1299, showed very low seed fertility and that the hybrid between W1297 and W1299 showed seed fertility comparable to parental lines. It suggests that W1299 also does not carry the egg-killer allele at *S57*. Additionally, W1297, W1299 and Jpn2 have originated from different places in Australia: W1297and W1299 are from the Northern Territory, and Jpn2 is from Queensland. According to Juliano et al. [43], most crosses between Northern Territory and Queensland accessions produced sterile hybrids, and DNA marker-based analyzes showed *O. meridionalis* genetic differentiation corresponding to geographic origin. In *Oryza sativa*, the carriers of the egg-killer allele and those of the abortive allele at *S5* corresponded to *indica* and *japonica* [8]. The distribution of alleles of *S57* in the wild populations of *O. meridiolnais* will test the role of the egg-killer gene in population differentiation in nature. 

## 4. Materials and Methods 

### 4.1. Plant Material

Three wild rice strains, Jpn1, Jpn2 and W1297, and one cultivated rice cultivar, Taichung 65 (T65) were used in this study. Detailed information on the three strains and the procedure of backcrossing has been already described in [24]. The number of plants in BC_2_F_1_ and BC_3_F_1_ generation in Figure 4 was larger than [24] because spare plants in each generation were planted and checked for seed fertility as T65 was used as a pollen donor. The year of cultivation, sowing date and transplanting date are shown in Table 7. Plant cultivation conditions followed Toyomoto et al. [24]: Germinated seeds were sown in nursery beds in a greenhouse. About two weeks after sowing, seedlings were transferred out of the greenhouse. Then, seedlings were planted in a paddy field at the Experimental Farm of Kagoshima University, Kagoshima, Japan.

### 4.2. Trait Evaluation

The recording of the heading date and evaluation of seed fertility and pollen fertility followed Toyomoto et al. [24]. 

### 4.3. DNA Analysis

DNA from leaves was extracted according to Toyomoto et al. [24]. PCR mixture, cycle, electrophoresis, DNA staining and gel image documentation also followed Ichitani et al. [44]. The linkage map was constructed using Antmap [45]. Kosambi function was adopted to calculate map distance.

### 4.4. DNA Markers

For the rough mapping of the causal gene of hybrid sterility, 76 PCR-based DNA markers covering the whole chromosomes in K.I.’s lab were examined (data not shown).

For the linkage analysis of the *S57* gene, new PCR-based DNA markers were designed because no available polymorphic DNA markers were closely located to S13374. Our strategy for designing co-dominant DNA markers was that using Nipponbare genome (GCF_001433945.1) [35] as a reference, insertion/deletion (indel) ranging from 5 to 100 base pairs shared by the two *O. meridionalis* accessions, W2112 (sequence information ID: SAMN02870169) [19] and IRGC105298 (SAMN03263073) [18], and not shared by the following AA genome species sequences was selected: *O. barthii* accession IRGC105608 (GCA_000182155.4) [19], *O. glumaepatula* accession GEN1233_2 ((SAMN02941241) [19], *O. nivara* accession IRGC100897 (SAMN02941243) [18], *O. rufipogon* accession IRGC100897 (SAMN02941243) [19], *O. rufipogon* accession W1943 (SAMEA1523624) [18], *O. longistaminata* accession W1413 (SAMD00003589) [46], *O. longistaminata* accession W1508 (SAMD00003590) [46], *O. sativa* cultivar Taicung 65 (SAMN03463201) [47] with the aid of TASUKE+ (https://ricegenome.dna.affrc.go.jp/ accessed online 15 December 2020) [48]. The selected indels were screened based on sequence similarity surrounding indels between genome assemblies of Nipponbare (Os-Nipponbare-Reference-IRGSP-1.0) and the *O. meridionalis* W2112 accession (GCA_000338895.3). The primer design followed Busung et al. [49]. The mismatch in primer sequences and genome sequences were checked using Oryzabase BLAST (https://shigen.nig.ac.jp/rice/oryzabase/locale/change?lang=en accessed online 15 December 2020) [31] and MBKBASE BLAST (http://mbkbase.org/rice/blastOL accessed online 15 December 2020) [50]. One base mismatch inside the primer sequence could be permitted, and we obtained enough PCR product.

## 5. Conclusions

A new egg-killer gene *S57* was detected in the cross between *O. sativa* and *O. meridionalis*. This gene was located on the distal end of the short arm of chromosome 1. An *O. meridionalis* strain Jpn2 carried an egg-killer allele, and *O. sativa* cultivar Taichung 65 carried an abortive allele. Fine mapping of this gene will test the universality of killer-protector systems reported before. Not all *O. meridionalis* strains carry the egg-killer allele. Analysis of the distribution of alleles of *S57* in the wild populations of *O. meridiolnais* will test the role of the egg-killer gene in population differentiation in nature.

## Figures and Tables

**Figure 1 plants-11-00378-f001:**
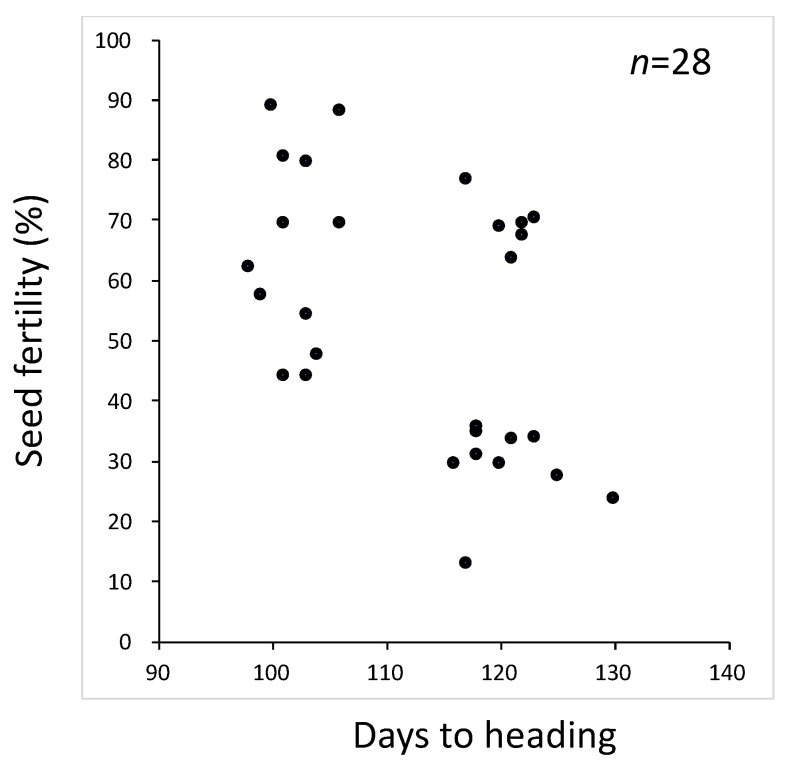
Scatter diagram of days to heading and seed fertility in the BC_5_F_2_ population using T65 as the recurrent parent and Jpn2 as the donor parent.

**Figure 2 plants-11-00378-f002:**
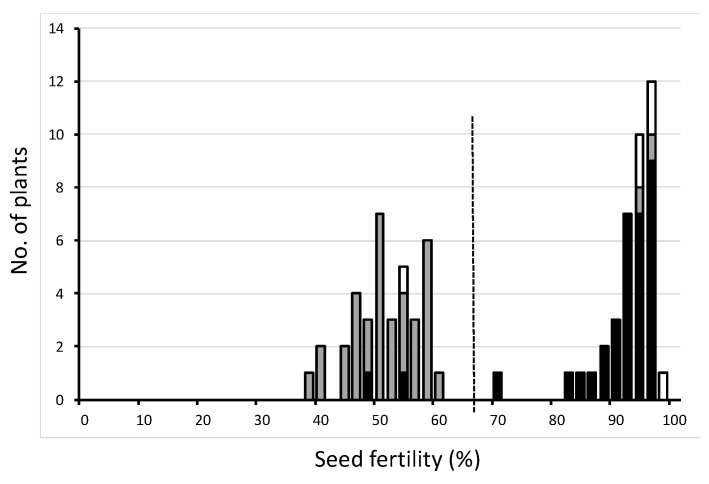
Frequency distribution of seed fertility of one BC_5_F_4_ population using T65 as the recurrent parent and Jpn2 as the donor parent. Three classified genotypes were assessed for S13374 as indicated: white, homozygous for T65; gray, heterozygous; black, homozygous for Jpn2.

**Figure 3 plants-11-00378-f003:**
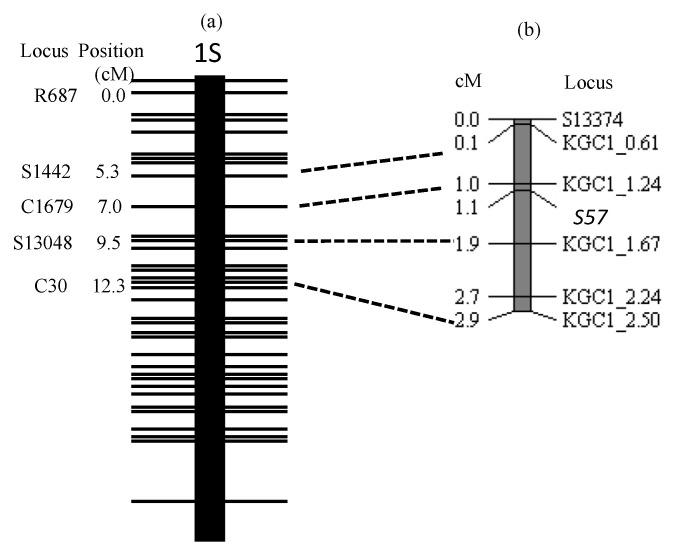
Linkage map showing *S57* gene on the short arm of chromosome 1. (**a**) RFLP framework map of the short arm of chromosome 1 modified from Harushima et al. [26]. (**b**) linkage map of *S57* gene constructed from the BC_5_F_5_ population using Jpn2 as donor parent and T65 as the recurrent parent (*n* = 506). DNA markers near each other on Nipponbare pseudomolecules are connected by dotted lines.

**Figure 4 plants-11-00378-f004:**
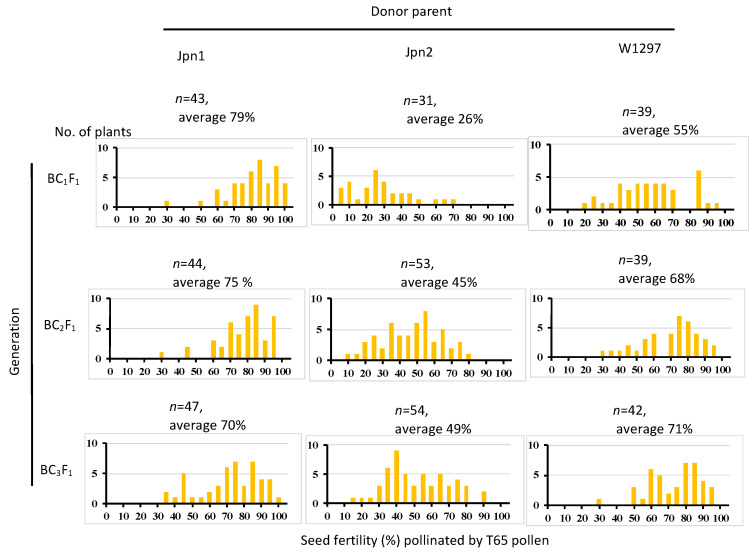
Seed fertility pollinated by T65 of backcross generation using Jpn1, Jpn2 and W1297 as donor parents and T65 as the recurrent parent.

**Figure 5 plants-11-00378-f005:**
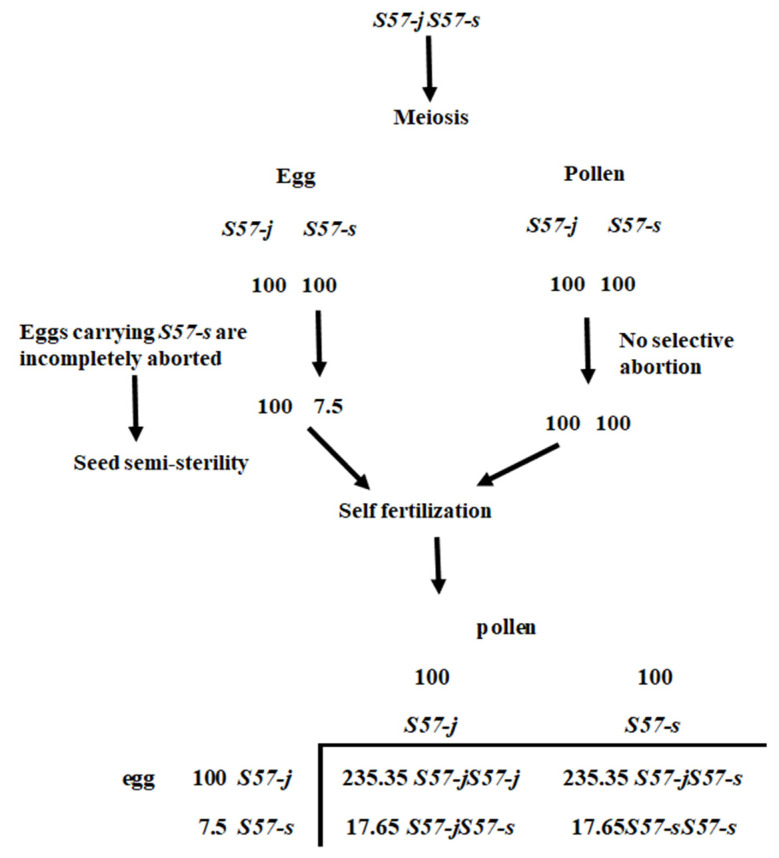
Genetic model explaining the seed semi-sterility and segregation distortion in the genotype of *S57-j S57-s* at *S57* locus. The transmission ratio of *S57-s* is inferred from observed data in Table 5.

**Table 1 plants-11-00378-t001:** Frequency distribution of seed fertility in BC_5_F_3_ generation using T65 as the recurrent parent and Jpn2 as the donor parent.

	Data in BC_5_F_2_																							
BC_5_F_3_ Line No.	Days to Heading	Seed Fertility	Genotype of S13374 ^1^																					
				Distribution of Seed Fertility (%) in BC_5_F_3_																				
																								Average
				10	15	20	25	30	35	40	45	50	55	60	65	70	75	80	85	90	95	100	Sum	
1	100	88.7	J					1				1				1	2		2	6	12	5	30	85.8
2	106	88.0	J															3	3	5	7	2	20	88.2
3	101	80.3	J											1		1		1	4	7	11	2	27	87.7
4	103	79.3	J		1	1	1		1	1				1			1	1	1	1	2		12	55.3
5	98	62.0	H			1	1	1	2	2	2	2	1	5		2	2		1	2	4	1	29	59.2
6	99	57.3	H				1	1			1	2	5	4	1	2	1	1	1	6	3		29	65.7
7	103	54.0	H								3	1	3	5							2		14	57.1
8	104	47.3	H								1	6	4	4	1				1	10	6	1	34	71.9
9	101	44.0	H								1	2	4	2	1	1			5	3			19	67.1
10	103	44.0	H									5	1	3	1	1	1	4	3	3	1		23	69.1

^1^ J and H respectively denote homozygotes for Jpn2 and heterozygotes.

**Table 2 plants-11-00378-t002:** Primer sequences of Indel markers designed or redesigned for mapping the *S57* gene.

Marker			Location on IRGSP 1.0 Pseudomolecule Chromosome 1		
Name		Primer Sequence	From	To	Source
S13374	F	GAGGCCGGTCAAGTTCCAC	456990	457233	RGP ^1^
	R	GATCTGAAAGCAGAGATAGATGATGC			redesigned
KGC1_0.61	F	GCTAGAATTAAGTAGCCACCAGCAG	612601	612727	This study
	R	AAACGGGCCATCGATCTCT			
KGC1_1.24	F	CACGTGTTATGTTGCTTGTTTGTAG	1240856	1240979	This study
	R	TACTGATCATTTCACCAAAAGTGAG			
KGC1_1.67	F	GACGTAGGACCTTCATCAGAGTAGAAT	1671591	1671774	This study
	R	AGCCGTACACCAGCGTCAG			
KGC1_2.24	F	ATCTCTCGATCGTTGATGTCATAAG	2244186	2244273	This study
	R	CTACTGTCAACCATCCCCTTTCT			
KGC1_2.50	F	ATCTGCTTGCCCTTACAATTACATA	2502301	2502480	This study
	R	TAAAGAGCTTTCACACAACATACCA			

^1^ RGP stands for Rice Genome Research Program. 332 PCR-based genetic markers on rice chromosomes. Available online: https://rgp.dna.affrc.go.jp/E/publicdata/caps/index.html (accessed on 30 December 2021).

**Table 3 plants-11-00378-t003:** Genotypes of selected BC_5_F_4_ and seed fertility of BC_5_F_5_ generation classified by KGC1_1.24.

	Seed																				
Line Number	Fertility of BC_5_F_4_	Genotype of DNA Marker ^1^																			
			KG	KG	KG	KG	KG		Seed Fertility (%) of BC_5_F_5_ Generation												
			C1_	C1_	C1_	C1_	C1_		Classified by the Genotype of KGC1_1.24												
		S13374	0.61	1.24	1.67	2.24	2.50		45	50	55	60	65	70	75	80	85	90	95	100	Sum
1	0.50	H	H	H	H	H	J	H		1	2	3									6
								J									1	2	6		9
2	0.50	H	H	H	H	H	H	H		1	2	3	1	1							8
								J										3	4		7
3	0.50	H	H	H	H	H	H	H				1	2	4							7
								J									1	2	4	1	8
4	0.50	H	H	H	H	H	H	H		1		2	5	2	1						11
								J										1	3		4
5	0.51	H	H	H	H	H	H	H				4	3	1							8
								J										1	5		6
6	0.51	H	H	H	H	H	H	H				3	1	4							8
								J										1	5	1	7
7	0.51	H	H	H	H	H	H	H				1	4	2	2						9
								J										3	1	2	6
8	0.53	H	H	H	H	H	H	H			1	4	4	3							12
								J											3		3
9	0.53	H	H	H	H	H	H	H		1	2	3	2								8
								J										2	4	1	7
10	0.93	H	H	J	J	J	J	J										5	8	2	15
11	0.95	H	H	J	J	J	J	J								1		6	6	2	15
13	0.48	J	H	H	H	H	H	H		2	1	4	1								8
								J									1	2	3	1	7
14	0.53	J	H	H	H	H	T	H			4	2	1	1							8
								J										1	4		5
15	0.53	T	T	H	H	H	H	H			3	3	1								7
								J								1		2	3	1	7
16	0.93	T	T	T	T	T	T	T						1				6	6	1	14
17	0.93	T	T	T	T	T	T	T								1	1	7	6		15
18	0.95	T	T	T	T	T	T	T									1	2	10	2	15
19	0.95	T	T	T	T	H	H	T			1						2	2	9	1	15
20	0.97	T	T	T	T	T	T	T										7	7	1	15

^1^ J, H and T respectively denote homozygotes for Jpn2, heterozygotes and homozygotes for T65.

**Table 4 plants-11-00378-t004:** Segregation of DNA markers of the progeny of the recombinants between KGC1_1.24 and KGC1_1.67 in BC_5_F_5_ and BC_5_F_6_ generation.

BC_5_F_5_								BC_5_F_6_																				
		Genotype at DNA Marker ^1^						DNA Marker Segregation at Seedling Stage																				
	Indi-	KG	KG	KG	KG	Seed		DNA		Genotype ^1^																		
Line	vidual	C1_	C1_	C1_	C1_	Fertility		Marker								Genotype ^1^	Seed Fertility (%)											
No.	No.	0.61	1.24	1.67	2.24			KGC1_	J	H	T	*χ^2^*(1:2:1)	*p*				35	40	45	50	55	60	-	80	90	95	100	Sum
2	74	H	H	T	T	0.64		1.24	12	30	4	7.043	0.030	*														
3	21	H	H	J	J	0.59		1.24	16	27	4	7.170	0.028	*		J									1	2	5	8
																H		3	1	8	2							14
																T											3	3
3	54	H	H	T	T	0.63		1.24	20	18	8	8.435	0.015	**														
7	61	H	H	T	T	0.93		1.24	12	19	11	0.429	0.807			J									1	2	6	9
																H										6	10	16
																T										4	1	5
8	71	J	J	H	H	0.91		1.67	8	27	12	1.723	0.422															
3	57	J	J	H	H	0.9		1.67	14	20	13	1.085	0.581			J										3	7	10
																H										6	6	12
																T									1		4	5
4	68	J	J	H	H	0.83		1.67	10	24	8	1.048	0.592															
5	1	J	J	H	H	0.95		1.67	12	25	9	0.739	0.691															
9	64	J	J	H	H	0.91		1.67	8	25	13	1.435	0.488															

^1^ J, H and T respectively denote homozygotes for Jpn2, heterozygotes and homozygotes for T65. * *p* < 0.05, ** *p* < 0.01.

**Table 5 plants-11-00378-t005:** Haplotypes around the segregation distortion region on chromosome 1 of BC_5_F_5_ using Jpn2 as the donor parent and T65 as the recurrent parent.

	Genotype ^1^			No. of
Haplotype	KGC1_1.24	*S57*	KGC1_1.67	Plants
1	J	J	J	230
2	H	H	H	251
3	T	T	T	16
4	H	H	J	1
5	H	H	T	1
6	H	H	T	1
7	J	J	H	5
8	H	T	T	1
			Sum	506

^1^ J, H and T respectively denote homozygotes for Jpn2, heterozygotes and homozygotes for T65.

**Table 6 plants-11-00378-t006:** Segregation of progeny of BC_5_F_5_ heterozygous for KGC1_1.24 genotype.

BC_5_F_5_		Genotype of the KGC1_1.24 ^1^											
Individual		BC_5_F_6_					BC_6_F_1_						
Number							T65 as Pollen Donor				T65 as Egg Donor		
					*P*				*P*				*P*
					χ^2^				χ^2^				χ^2^
		J	H	T	(1:2:1)		H	T	(1:1)		H	T	(1:1)
T1		21	20	4	0.001		19	1	0.001>		18	16	0.732
T2		18	26	2	0.003		32	6	0.001>		20	26	0.476
T3		20	22	2	0.001		30	5	0.001>		25	19	0.366
Sum		59	68	8	0.001>		81	12	0.001>		63	61	0.857

^1^ J, H and T respectively denote homozygotes for Jpn2, heterozygotes and homozygotes for T65.

**Table 7 plants-11-00378-t007:** Year of cultivation, sowing date and transplanting date of the materials in this study.

Generation		Year	Sowing Date	Transplanting Date
BC_1_F_1_		2012	May 27	July 3
BC_2_F_1_		2013	May 24	July 4
BC_3_F_1_		2014	May 24	July 1
BC_5_F_2_		2017	May 25	July 4
BC_5_F_3_		2018	June 1	July 6
BC_5_F_4_		2019	June 5	July 3
BC_5_F_5_		2020	May 28	July 1
BC_5_F_6_	BC_6_F_1_	2021	May 22	July 16

## Data Availability

Data is contained within the article.

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
