# Peer review of "New Hybrid Spikelet Sterility Gene Found in Interspecific Cross between *Oryza sativa* and *O. meridionalis"

_plants, 2022, doi:10.3390/plants11030378_

Round 1
Reviewer 1 Report
The article is interesting, dealing with a crucial topic of plant breeding, i. e. hybridization, in a very important crop like rice. However, I think that there are some minor drawbacks that must be faced. Although the abstract is good giving sufficient information of the article it lacks in giving clear conclusions or suggestions. The beginning of introduction is similar with the beginning of the abstract and this has also to be faced. Furthermore, introduction is very short and has to be improved and contain some information and on safety issues of rice–hybrid cropping. The authors must remember the serious problems caused by cropping rice hybrids in India some years ago and ensure their safe use.
The results of the article gave me the impression that I was reading the discussion and not a reference of the results found with this study. Furthermore, I had the same problem and reading discussion. The last must be enriched with more references, supporting or not the results of the study. Finally, the most crucial drawback was the lack of clear conclusions which have to be included in the article. All tables and figures are good and the references used are current.
Author Response
We gratefully acknowledge the editor and the two reviewers for valuable comments. Our replies to the comments are presented below. We add many sentences, folloing the comments by Reviewer 1. Using track system might make the manuscript difficult to read. We highlighted the added sentences and references with yellow background. The correction following the comments by reviewer 2 was written in red.
Comments by Reviewer 1
The article is interesting, dealing with a crucial topic of plant breeding, i. e. hybridization, in a very important crop like rice. However, I think that there are some minor drawbacks that must be faced. Although the abstract is good giving sufficient information of the article it lacks in giving clear conclusions or suggestions. The beginning of introduction is similar with the beginning of the abstract and this has also to be faced. Furthermore, introduction is very short and has to be improved and contain some information and on safety issues of rice–hybrid cropping. The authors must remember the serious problems caused by cropping rice hybrids in India some years ago and ensure their safe use.
The results of the article gave me the impression that I was reading the discussion and not a reference of the results found with this study. Furthermore, I had the same problem and reading discussion. The last must be enriched with more references, supporting or not the results of the study. Finally, the most crucial drawback was the lack of clear conclusions which have to be included in the article. All tables and figures are good and the references used are current.
Response: We thank you for your valuable comments.
We could not find problems caused by cropping rice hybrids in India some years ago. Instead, We added some sentences explaining that the genetics of hybrid sterility could contribute to hybrid rice breeding exploiting hybrid vigor, and prior knowledge about hybrid sterility between O. sativaand O. meridionalisin the introduction section.
In the result section, we removed some redundant sentences, and merged Table 4 and Table 5 into a new Table 4 to focus on the important points of this study.
In the discussion section, we added some sentences describing the recent accomplishment of understanding the genetics of hybrid sterility genes, future prospects of studying the S57 gene and the possible contribution of the S57 gene to the genetics of the hybrid sterility genes and O. meridionalis differentiation.
We added a conclusion section to make the conclusion of this paper clear.
Reviewer 2 Report
Review on plants-1559907
New Hybrid Spikelet Sterility Gene Found in Interspecific Cross between Oryza Sativa and O. Meridionalis
By Katsuyuki Ichitani, Daiki Toyomoto, Masato Uemoura, Kentaro Monda, Makoto Ichikawa, Robert Henry, Tadashi Sato, Satoru Taura and Ryuji Ishikawa
The Authors present an interesting, well-written and concretely illustrated study which advances rice breeding. This manuscript is suggestive for publication in MDPI Plants.
There are only a few detailed comments to be considered before “printing”.
[line, comment]
44, models
50, 1:1
Figure 1, y-axis and x-axis numbering, the question marks are probably incorrect
120, 37:39
120, 1:1
123, by that.
145, 1.67 (
146, 3).
147, line 7 individual
155, generation.
161, as shown in Table 6.
164, 3).
Table 5, 1st line, Parental
196, 1:1
Table 7, Heading, The corresponding statistical method and meaning of P and chi should be mentioned.
Figure 4, The alignment of boxes and letters should be optimized.
279-285, The conditions under which plants have been cultivated should be mentioned (greenhouse/laboratory/field, temperature, humidity, light, etc.).
300, using Nipponbare
302, Italics: O. meridionalis
Author Response
We gratefully acknowledge the editor and the two reviewers for valuable comments. Our replies to the comments are presented below. We add many sentences, folloing the comments by Reviewer 1. Using track system might make the manuscript difficult to read. We highlighted the added sentences and references with yellow background. The correction following the comments by reviewer 2 was written in red.
Comments by Reviewer 2
There are only a few detailed comments to be considered before “printing”.
[line, comment]
44, models
Response: We revised the Introduction, following the comments by Reviewer 1, and the original sentence was changed.
50, 1:1
Response: We revised as you suggested.
Figure 1, y-axis and x-axis numbering, the question marks are probably incorrect.
Response: We are sorry for the mistake. we corrected it.
120, 37:39
Response: We revised as you suggested.
120, 1:1
Response: We revised as you suggested.
123, by that.
Response: We revised the Results, following the comments by Reviewer 1, and the original sentence containing "by that" was removed.
145, 1.67 (
Response: We revised as you suggested.
146, 3).
Response: We revised as you suggested. And Table 4 is correct.
147, line 7 individual
Response: We are sorry for the mistake. We corrected it.
155, generation.
Response: We revised as you suggested.
161, as shown in Table 6.
Response: We revised as you suggested.
164, 3).
Response: We revised as you suggested.
Table 5, 1st line, Parental
Response: We merged Table 4 and Table 5 into a new Table 4 so that the relationship between BC5F5 and BC5F6 could be easily understood.
196, 1:1
Response: We revised as you suggested.
Table 7, Heading, The corresponding statistical method and meaning of P and chi should be mentioned.
Response: We are sorry for the mistake. We corrected it.
Figure 4, The alignment of boxes and letters should be optimized.
Response: We revised as you suggested.
279-285, The conditions under which plants have been cultivated should be mentioned (greenhouse/laboratory/field, temperature, humidity, light, etc.).
Response: We revised as you suggested.
300, using Nipponbare
Response: We are sorry for the mistake. We corrected it.
302, Italics: O. meridionalis
Response: We are sorry for the mistake. We corrected it.